# Analysis of Risk Factors for Tracheal Stenosis Managed during COVID-19 Pandemic: A Retrospective, Case-Control Study from Two European Referral Centre

**DOI:** 10.3390/jpm13050729

**Published:** 2023-04-26

**Authors:** Giuseppe Mangiameli, Gianluca Perroni, Andrea Costantino, Armando De Virgilio, Luca Malvezzi, Giuseppe Mercante, Veronica Maria Giudici, Giorgio Maria Ferraroli, Emanuele Voulaz, Caterina Giannitto, Fabio Acocella, Ilaria Onorati, Emmanuel Martinod, Umberto Cariboni

**Affiliations:** 1Division of Thoracic Surgery, IRCCS Humanitas Research Hospital, Via Manzoni 56, 20089 Rozzano, MI, Italy; 2Department of Biomedical Sciences, Humanitas University, Via Rita Levi Montalcini 4, 20072 Pieve Emanuele, MI, Italy; 3Otorhinolaryngology, Head & Neck Surgery Unit, IRCCS Humanitas Research Hospital, Via Manzoni 56, 20089 Rozzano, MI, Italy; 4Department of Diagnostic Radiology, IRCCS Humanitas Research Hospital, Via Manzoni 56, 20089 Rozzano, MI, Italy; 5Department of Veterinary Medicine and Animal Science, University of Milan, Via dell’Università 6, 26900 Lodi, LO, Italy; 6Assistance Publique—Hôpitaux de Paris (AP-HP), Hôpitaux Universitaires Paris Seine-Saint-Denis, Hôpital Avicenne, Chirurgie Thoracique et Vasculaire, Université Sorbonne Paris Nord, Faculté de Médecine SMBH, Bobigny, 93000 Paris, France

**Keywords:** tracheal stenosis, tracheal surgery, COVID-related tracheal stenosis

## Abstract

Introduction: Benign subglottic/tracheal stenosis (SG/TS) is a life-threatening condition commonly caused by prolonged endotracheal intubation or tracheostomy. Invasive mechanical ventilation was frequently used to manage severe COVID-19, resulting in an increased number of patients with various degrees of residual stenosis following respiratory weaning. The aim of this study was to compare demographics, radiological characteristics, and surgical outcomes between COVID-19 and non-COVID patients treated for tracheal stenosis and investigate the potential differences between the groups. Materials and methods: We retrospectively retrieved electronical medical records of patients managed at two referral centers for airways diseases (IRCCS Humanitas Research Hospital and Avicenne Hospital) with tracheal stenosis between March 2020 and May 2022 and grouped according to SAR-CoV-2 infection status. All patients underwent a radiological and endoscopic evaluation followed by multidisciplinary team consultation. Follow-up was performed through quarterly outpatient consultation. Clinical findings and outcomes were analyzed by using SPPS software. A significance level of 5% (*p* < 0.05) was adopted for comparisons. Results: A total of 59 patients with a mean age of 56.4 (±13.4) years were surgically managed. Tracheal stenosis was COVID related in 36 (61%) patients. Obesity was frequent in the COVID-19 group (29.7 ± 5.4 vs. 26.9 ± 3, *p* = 0.043) while no difference was found regarding age, sex, number, and types of comorbidities between the two groups. In the COVID-19 group, orotracheal intubation lasted longer (17.7 ± 14.5 vs. 9.7 ± 5.8 days, *p* = 0.001), tracheotomy (80%, *p* = 0.003) as well as re-tracheotomy (6% of cases, *p* = 0.025) were more frequent and tracheotomy maintenance was longer (21.5 ± 11.9 days, *p* = 0.006) when compared to the non-COVID group. COVID-19 stenosis was located more distal from vocal folds (3.0 ± 1.86 vs. 1.8 ± 2.03 cm) yet without evidence of a difference (*p* = 0.07). The number of tracheal rings involved was lower in the non-COVID group (1.7 ± 1 vs. 2.6 ± 0.8 *p* = 0.001) and stenosis were more frequently managed by rigid bronchoscopy (74% vs. 47%, *p* = 0.04) when compared to the COVID-19 group. Finally, no difference in recurrence rate was detected between the groups (35% vs. 15%, *p* = 0.18). Conclusions: Obesity, a longer time of intubation, tracheostomy, re-tracheostomy, and longer decannulation time occurred more frequently in COVID-related tracheal stenosis. These events may explain the higher number of tracheal rings involved, although we cannot exclude the direct role of SARS-CoV-2 infection in the genesis of tracheal stenosis. Further studies with in vitro/in vivo models will be helpful to better understand the role of inflammatory status caused by SARS-CoV-2 in upper airways.

## 1. Introduction

Benign subglottic/tracheal stenosis (SG/TS) is a debilitating and potentially fatal condition caused by iatrogenic events during endotracheal intubation or tracheostomy [1,2]. Grillo identified high-volume, high-pressure cuffs as the primary cause of injuries leading to post-intubation tracheal stenosis in 1966 [3,4].

Since March 2020, the novel Coronavirus disease 2019 (COVID-19), caused by the severe acute respiratory syndrome coronavirus 2 (SARS-CoV-2), has spread rapidly becoming a major global public health emergency. Since the onset of the SARS-CoV-2 pandemic, an increase in the occurrence of benign subglottic/tracheal stenosis (SG/TS) has been noted, most likely as a result of several factors, including the large number of critically ill patients who required endotracheal intubation, the prolonged mechanical ventilation time for a median of 18 days; the high rate of re-intubation, the large number of tracheostomies performed; the inflammatory state of the upper airways linked to the viral infection [5]. Thus, in the last two years, an increased number of patients with tracheal stenosis, as well as patients with unrelated tracheal stenosis, have been referred to referral centers for airway surgery [6].

We present a retrospective review of all patients treated for tracheal stenosis in two European referral centers during the SARS-CoV-2 pandemic (IRCCS Humanitas Research Hospital, Rozzano, Italy, and Avicenne Hospital, Bobigny, France) from March 2020 to May 2022. The purpose of this study was to investigate and compare demographics, radiological findings, type of treatment used, rate of recurrence, and risk factors of tracheal stenosis occurred in the general population and SARS-CoV-2 patients.

## 2. Material and Methods

### 2.1. Study Population

A retrospective review of all patients affected by tracheal stenosis receiving endoscopic and/or surgical treatment from March 2020 until May 2022 in two division of thoracic surgery (IRCCS Humanitas Research Hospital, Rozzano, Italy, and Avicenne Hospital, Bobigny, France) were performed. All patients with suspected or documented tracheal stenosis were evaluated with computed tomography (CT) of the neck and chest and flexible fiberoptic bronchoscopy. This study was approved by the Ethics Committee of Avicenne Hospital, “CLEA”, on 15 November 2021 (ID number CLEA-2021-224) and informed consent was obtained for each patient. 

Ethical approval was waived by the local Ethics Committee of Humanitas University in view of the retrospective nature of the study and all the procedures being performed were part of the routine care and adhere to the tenets of the Declaration of Helsinki.

### 2.2. Demographic Data

Baseline characteristics of patients were collected upon admission to the weaning center, including demographics, clinical features, and comorbidities. Patient data on tracheal stenosis diagnosis (idiopathic or post-intubation/tracheostomy), COVID infection, length of intubation with invasive mechanical ventilation (days), type of performed tracheostomy (percutaneous vs. surgical), decannulation time, and the occurrence of redo-tracheostomy and its duration were also recorded. 

### 2.3. Radiological Data

All tracheal CT scans were performed with a General Electric CT Scanner (HiSpeed CT/i, GE, Milwaukee, WI, USA) with standard reconstruction software. The protocol for CT scanning required 3 mm transaxial collimation beginning at the level of the hyoid bone and extending to the carina. A 20 cm field of view was used with standard interpolation and algorithm. Pitch was set at 1.0 to 1.5 with a 1.5 mm overlap. Coronal and sagittal reconstructions were performed at 1.5 mm using shaded surface reconstruction. The degree of tracheal stenosis was determined by measuring the greatest percentage of narrowing in the transaxial plane and corroborating these findings with coronal and sagittal tracheal reconstructions. Tracheal stenosis was defined as greater than 10% reduction in the transaxial tracheal diameter. If coronal or sagittal reconstructions indicated a stenosis greater than that found on transaxial views, the larger percentage stenosis was recorded. Tracheal stenosis was classified according to their severity in Grade 1 with a luminal narrowing < 50%, Grade 2 with luminal narrowing ≥ 50% but <70%, and Grade 3 with a luminal narrowing ≥ 70% according to the Myer-Cotton grading system [7].

To determine the stenosis length and length from the vocal cord: we used the same methods proposed by Kamal et al. [8], who used the program ruler for measuring the maximal projectional length of the tracheal luminal narrowing. Radiological data were collected by one of the authors (C.G) who had not been involved in surgical management and had no knowledge of the patients’ clinical symptoms. 

### 2.4. Flexible Bronchoscopy

Flexible bronchoscopy was systematically performed under topical anesthesia with 4% lidocaine and vigilant sedation by using midazolam 2 mg e.v. A fiberoptic laryngoscope (Fujifilm, Tokyo, Japan) was passed nasally to evaluate the vocal cords and the posterior pharyngeal and subglottic areas. Any abnormalities identified were recorded. We describe the type, location, and severity of tracheal stenosis detected through flexible bronchoscopy. The tracheal stenosis was classified as eccentric or concentric. All procedures were performed by an expert thoracic surgeon (G.F and I.O).

### 2.5. Rigid Bronchoscopy

Rigid bronchoscopy procedures were realized according to our protocol: patients were pre-oxygenated by high flow nasal-cannula oxygen therapy (HFNO) at 100% fractional inspired oxygen (FiO_2_) administration; after general anesthesia by target-controlled infusion of propofol/remifentanil, patients were intubated thought rigid bronchoscope and ventilation was assured manually during the non-interventional time.

The endoscopic procedures performed depend on the type of laryngotracheal lesion, and they consisted of tracheal dilatation by tracheobronchial balloon (Pulmonary BostonR) +/− injection of corticosteroid or mitomycin, resection of granulomas, thermocoagulation, placement of the tracheal stent, and biopsy of abnormal tracheal tissue for histological examination. All procedures were performed by an expert thoracic surgeon (G.F and I.O).

### 2.6. Surgical Treatment

Definitive surgical treatment consisted of resection of the stenotic segment of airway with primary end-to-end anastomosis for reconstruction through cervicotomy. During surgery ventilation was assured through cross-field ventilation or by the placement of a laryngeal mask according to the patient. The end-to-end anastomosis was performed with interrupted, lubricated, 4–0 polyglactin 910 (Vicryl; Ethicon, Sommerville, NJ, USA). No muscle flap was used to cover tracheal anastomosis. The number of tracheal rings involved was systematically recorded. All procedures were performed by an expert thoracic surgeon (U.C and E.M). In Figure 1 are shown two tracheal resections, respectively, performed for non-COVID (A) and COVID stenosis (B).

### 2.7. Post-Procedural Complications and Follow-Up

The postoperative complications were scored according to Clavien–Dindo criteria [9]. Perioperative mortality was defined as death within 30 days of surgery or the same hospital stay. After discharge, each patient was scheduled for a flexible bronchoscopy after 1 month. Successively, clinical data regarding follow-up were collected through quarterly outpatient consultations.

### 2.8. Statistical Analysis

All data were collected and stored in a Microsoft Excel^®^ spreadsheet. Dichotomous variables were reported as counts and percentage, while continuous variables as mean and standard deviation (SD), or as median if the values were not normally distributed in the Shapiro–Wilk normality test. The 2-sample Wilcoxon rank-sum test was used to test the hypothesis of equal distributions among the COVID and non-COVID groups for continuous variables. For categorical variables, the proportions in the 2 groups were compared using the χ2 or Fisher’s exact test, as appropriate. Statistical analyses were performed using the R software for statistical computing (R version 4.0.1). A value of *p* < 0.05 was considered to indicate statistical significance.

## 3. Results

### 3.1. Demographic Data

A total of 59 patients with a mean age of 56.4 (±13.4) years met the inclusion criteria, 27 (46%) were female. The COVID-19 group consisted of 36 patients (61%). The patients were all obese, with a median BMI of 28.6 (±4.8). BMI was higher in the COVID-19 group compared to the non-COVID group (29.7 ± 5.4 vs. 26.9 ± 3, *p* = 0.043), while age, gender distribution, number and types of comorbidities were lower. The smoking habits of the groups were similar.

Tracheal stenosis was idiopathic in 7 out of 23 patients in the non-COVID group (30%). A complete list of clinical characteristics can be found in Table 1. Orotracheal intubation was maintained for 17.7 days (±14.5) in the COVID patients, and this result was significantly higher when compared to non-COVID patients (9.7 ± 5.8 days, *p* = 0.001). When compared to non-COVID patients, tracheotomy was performed more frequently (80% of cases, *p* = 0.003), as was re-tracheotomy (6% of cases, *p* = 0.025), and tracheostomy tube maintenance was longer (21.5 ± 11.9 days, *p* = 0.006) (see Table 1).

### 3.2. Radiological Findings

Tracheal stenosis occurred more frequently in the subglottic area (57%), was grade 3–4 in the majority of cases (61.9%) and was located 2.5 (±1.9) cm below the vocal cords. When compared to the non-COVID group, stenosis occurred frequently at 3 (±1.86) cm below the vocal cord (*p* = 0.07). There was no difference between the two groups in terms of stenosis site as well as grade (Table 2).

### 3.3. Procedural Findings

Tracheal stenosis shape evaluated during flexible bronchoscopy was more frequently concentric (82% of cases) without statistically significant differences between the two groups. Patients underwent a median of 1.6 (±1.3) endoscopic procedures without differences between the two groups. Tracheal resection was performed in 32 (54%) patients with a trend towards significance (0.06) in the COVID-19 group. Tracheal stenosis submitted to surgical resection were more extended compared to non-COVID tracheal stenosis involving a larger number of tracheal rings (2.6 ± 0.8 vs. 1.7 ± 1.0, *p* = 0.001). Non-COVID tracheal stenosis was more commonly treated with rigid bronchoscopy and dilatation (with or without corticosteroid/mitomycin infiltration) (*p* 0.043). Although not statistically significant, tracheal stents were placed more frequently in non-COVID patients (27% vs. 7.4%). (Table 3).

### 3.4. Post-Procedural Complications and Follow Up

A total of nine complications (16%) occurred in the study population without 30-day mortality. Specifically, early postoperative complication rate was 16.6% and 13.0% in the COVID and non-COVID group, respectively. Each complication that occurred in the COVID and non-COVID group is summarized in Table 4.

In the first group we experienced six complications.

− The first patient was not extubated after surgery and was transferred to ICU, where after extubation, it was necessary to perform a surgical tracheostomy due to bilateral vocal fords paralysis in the adduction. It was decannulated after posterior cordectomy 9 months later. The patient is currently asymptomatic.− The second patient experienced postoperative pneumonia (Grade III), requiring orotracheal intubation for respiratory failure for four days. After extubation, he experienced postoperative dysphagia, needing a percutaneous gastrostomy. − The third patient experimented a clostridium difficile infection (grade II) successfully treated by antibiotic therapy. − The fourth patient experienced postoperative pneumonia (Grade III), requiring orotracheal intubation for respiratory failure for four days. After extubation, he experienced an acute kidney injury requiring dialysis for 3 days. − The fifth patient was not extubated after surgery and was transferred to ICU, where he developed VAP (postoperative grade III pneumonia), requiring 8 days of intubation.− The sixth patient was referred to our center, where a T tube was positioned. One week after, due to the occurrence of severe dyspnea, he was submitted to removal of the T tube and de-obstruction of the subglottal space due to the occurrence of granulomas. A Shinley tracheostomy was placed after the procedure. − In the non-COVID group three complications occurred:− The first patient experienced an anastomotic dehiscence of anterior wall of tracheal anastomosis on postoperative day 4. A definitive tracheostomy was performed.− In the second patient it was not possible to safely perform the anastomosis of the anterior wall due to frailty of residual tracheal rings. Thus, a definitive tracheostomy was performed. − The last one presented a severe subglottic stenosis needing the definitive placement of a T Montgomery Tube 

There was no statistically significant difference in recurrence between the COVID and non-COVID groups from the first diagnosis and regardless of treatment performed (Table 3).

## 4. Discussion

As a result of the COVID pandemic, a greater number of tracheal stenosis cases have been managed in referral airway centers over the last two years [10,11,12,13]. In this study, we compared the demographics, radiological characteristics, and surgical outcomes of COVID-19 and non-COVID patients treated for tracheal stenosis to spot any differences.

Obesity, a longer time of intubation, tracheostomy, re-tracheostomy, and a longer decannulation time were the identified risk factors that can play a role in the occurrence of COVID related tracheal stenosis.

Obesity, specifically a BMI ≥ 29, was significantly associated with the occurrence of tracheal stenosis in COVID patients in our study. This finding was previously reported by Martinod et al. in a recent study of nine patients with laryngotracheal post-intubation/tracheostomy stenosis following COVID-19 infection and a BMI greater than 30 [14]. Several authors have demonstrated the correlation between COVID-19 patients requiring invasive mechanical ventilation and obesity, which is also a proven risk factor for SG/TSs [15,16,17]. These studies confirmed a high frequency of obesity among patients admitted to intensive care for SARS-CoV-2 with a disease severity increased due to BMI. Consequently, it can be postulated that a certain amount of these patients will develop SG/T cicatricial concentric stenosis after extubation [18]. Obesity is usually combined in COVID patients with several other comorbidities such as advanced age, post-COVID-19 severe cardiopulmonary conditions or morbid ICU polyneuropathy. In this scenario, selected patients with extreme comorbidities, in whom extensive surgery and complex postoperative course might represent an overshooting, palliative care could be considered by positioning an endoluminal stent or, if present, leaving the tracheostomy in situ.

In our series, the time of orotracheal intubation was significantly longer in the COVID-19 group (17.7 days vs. 9.7), and the risk of developing tracheal stenosis increased by 20% per day of intubation regardless of COVID status. Reports from several studies have demonstrated that COVID-19 patients had a median duration of ventilation of 17 days and a high frequency of re-intubation [19,20]. In pre-COVID-19 settings, tracheostomy performed after 7–14 days from endotracheal intubation significantly improved the chance of successful weaning and lowered the risk of complications and mortality when compared to long-term maintenance of the orotracheal tube in place [21]. The immediate consequence of this data that can be applied to the next hypothetical pandemic is likely the early performance of tracheostomy, which would provide the benefits of faster weaning, while reducing the risk of tracheal stenosis development. In COVID patients, however, several additional factors, such as large caliber of tubes, over-cuffed intubation, and prone position ventilation, may contribute to the mechanism underlying the stenosis. Both of these variables were not investigated in our series.

For obvious reasons, use of large caliber tubes exposes the patient to an endo-laryngeal (especially at the level of the posterior commissure and subglottic) and/or endotracheal damage, whenever the intubation should be prolonged in time [22]. Moreover, it is well known that poor monitoring of tracheal tube cuff pressure, with undue over-inflation, may result in further ischemic damage to the airway mucosa. Careful use of a manometer is recommended to keep safe cuff pressure values between 20 and 30 cm H_2_O [23,24].

In critically ill patients affected by the COVID-19, early prone position (PP) has been commonly used to improve oxygenation and survival as compared to a supine position thanks to the recruitment of atelectatic dorsal lung areas and the redistribution of lung ventilation toward still well-perfused areas [25]. During PP repeated microtraumas could play a possible role in the pathogenesis of stenosis. However, to the best of our knowledge, no precise description of what happens to the laryngotracheal junction of COVID-19 patients ventilated for prolonged time in prone position has been so far produced.

In the current study, COVID patients were tracheotomized significantly more frequently than non-COVID patients. It is interesting to note that the site of stenosis in the COVID-19 group is at the site of tracheostomy (30.2 ± 20.3 mm from vocal folds). Furthermore, due to radiological artifacts, the measure of the distance of stenosis from vocal cords was not performed when a tracheostomy was in place. This could explain the trend toward significance (*p* = 0.07) between the two groups. 

However, the exact role of tracheostomy in determining tracheal stenosis is not clear. Even the type of tracheostomy, surgical or percutaneous, appears to be unrelated to an increased tendency to stenosis. Several studies report findings consistent with our results, concluding that the choice of these two techniques should not be based on the risk of tracheal stenosis [26,27]. 

The possible explication of this data is probably the fact that the clinical practice for COVID-19 patients admitted in several ICUs worldwide was to try to post-pone tracheostomy until the patient no longer needed to be ventilated in the prone position and was determined to be cleared of the virus with isolation precautions ceasing. Thus, when strictly followed, such a policy may mean that patients remained intubated for up to 3–4 weeks.

On the other side, in our series the mean time of intubation was relatively briefer than 3–4 weeks.

We have identified a great amount of tracheostomy, usually performed in peripheral centers, presenting surgical related problems such as higher position (just below cricoid or in the first tracheal ring) or characterized by the presence of broken tracheal rings and/or malacia at the level of the tracheostomy site or distal airway. We can speculate that the high rate of broken tracheal rings when performing tracheal resection in COVID-19 was probably a consequence of suboptimal conditions during pandemic such as: patient referred to non-teaching centers and high occurrence of emergency tracheostomy [28].

In our series, re-tracheostomy occurred exclusively in COVID patients, confirming it as a major risk factor for the development of tracheal stenosis. In most cases, tracheostomy was performed during the same hospitalization. Additionally, considering the close association with the occurrence of tracheal stenosis is crucial during the initial clinical evaluation and therapeutic management to avoid as much as possible a tracheostomy or redo-tracheostomy with further damage to the airway framework [28].

In this study, COVID tracheal stenoses were more likely to be treated by surgery compared to non-COVID considering their location, usually more distal from vocal folds, and their trend to involve a greater number of tracheal rings. The evidence of a higher number of tracheal rings involved in the COVID-19 group could explain the tendency to perform tracheal resection, although without a statistically significant difference when compared to the non-COVID group (*p* = 0.06). We can hypothesize an inflammatory role of SARS-CoV-2 in determining extensive tracheal ring involvement, although we cannot exclude a combination of mechanical factors (e.g., re-tracheostomy, longer decannulation time that were both more frequent in the COVID-19 group) as a main cause [29,30].

Our study presents several limits. First, the exact role of different drugs (corticosteroid and/or antiviral) used in treating COVID infection has not been investigated and their eventual involvement in inducing SG/TS. Second, the role of prone position in determining the occurrence of stenosis in COVID patients has not been investigated. 

Third, a lot of patients during the pandemic were hospitalized in peripheral centers with low experience in the management of airways. Thus, we have no knowledge about an important risk factor contributing to tracheal stenosis development, as difficult intubation is defined as repeated failing attempts to introduce an endotracheal tube, prolonged duration of such maneuvers, or necessity for multiple approaches and/or intubation devices. Probably in these cases, endoscopic removal of granulation tissue, dilatations by balloon, bougies or rigid bronchoscopy instrumentation, and steroid injection may represent the best temporary measure to obtain partial relief from dyspnea and to delay needing surgery after the resolution of local inflammatory status. If absolutely necessary, tracheostomy should be performed on the already damaged and stenosed cricotracheal tract to not further compromise the adjacent healthy tracheal rings.

## 5. Conclusions

Our series confirms that the COVID pandemic has increased the number of tracheal stenoses that are normally managed in referral centers. Obesity, longer time of intubation, tracheostomy, re-tracheostomy, and longer decannulation time are the major risk factors contributing to the development of tracheal stenosis in these patients. The treatment of these types of stenosis is usually complex, and it should be reserved for tertiary centers with specific expertise in proper evaluation and treatment of LTS

## Figures and Tables

**Figure 1 jpm-13-00729-f001:**
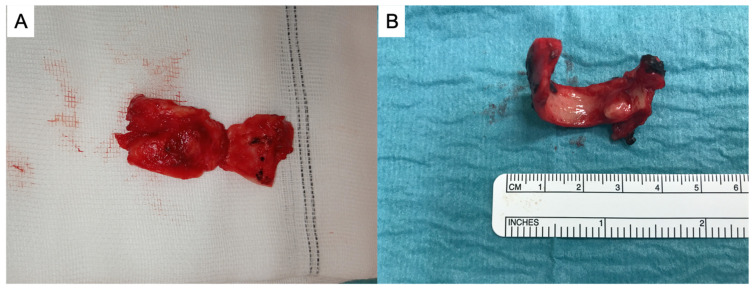
Tracheal resections for non-COVID stenosis (**A**) and for COVID-stenosis (**B**).

**Table 1 jpm-13-00729-t001:** Clinical characteristics and invasive airways management.

Variable		Overall *N* = 59	COVID Status		*p*-Value
			No, *N* = 23	Yes, *N* = 36	
Gender	Female	27 (46%)	13 (57%)	14 (39%)	0.18
	Male	32 (54%)	10 (43%)	22 (61%)	
Age (years)		56.4 (13.4)	57.9 (14.6)	55.4 (12.7)	0.37
Comorbidities		42 (71%)	16 (70%)	26 (72%)	0.83
Cardiologic		23 (39%)	8 (35%)	15 (42%)	0.6
Pneumological		12 (20%)	6 (26%)	6 (17%)	0.51
Diabetes		16 (27%)	4 (17%)	12 (33%)	0.18
BMI		28.6 (4.8)	26.9 (3.0)	29.7 (5.4)	0.043
BMI (dichotomized)	High	22 (37%)	4 (17%)	18 (50%)	0.012
	Low	37 (63%)	19 (83%)	18 (50%)	
Smoker		17 (29%)	9 (39%)	8 (22%)	0.16
Intubation (days)		14.6 (12.4)	9.7 (5.8)	17.7 (14.5)	0.001
Tracheotomy	Not performed	21 (36%)	14 (64%)	7 (19%)	0.003
	Percutaneous	16 (28%)	3 (14%)	13 (36%)	
	Surgical	21 (36%)	5 (23%)	16 (44%)	
Re-tracheotomy		6 (12%)	0 (0%)	6 (22%)	0.025
Decannulation (days)		16.5 (15.5)	10.9 (17.3)	21.5 (11.9)	0.006

**Table 2 jpm-13-00729-t002:** Radiological findings.

Variable		Overall, *N* = 59	COVID Status		*p*-Value
			No, *N* = 23	Yes, *N* = 36	
Site	Cervical trachea	18 (43%)	10 (50%)	8 (36%)	0.37
	Subglottic	24 (57%)	10 (50%)	14 (64%)	
Grade	1	3 (8.8%)	1 (7.1%)	2 (10%)	0.92
	2	10 (29%)	5 (36%)	5 (25%)	
	3	20 (59%)	8 (57%)	12 (60%)	
	4	1 (2.9%)	0 (0%)	1 (5.0%)	
Length (cm)		7.7 (4.8)	8.0 (4.7)	7.5 (4.9)	0.6
Distance (cm)		25.3 (19.9)	18.2 (20.3)	30.2 (18.6)	0.076

**Table 3 jpm-13-00729-t003:** Procedural findings and recurrence.

Variable		Overall, *N* = 59	Non-COVID *N* = 23	COVID-19 *N* = 36	*p*-Value
Shape	Concentric	40 (82%)	17 (77%)	23 (85%)	0.71
	Eccentric	9 (18%)	5 (23%)	4 (15%)	
N° of Procedures		1.6 (1.3)	1.4 (0.7)	1.6 (1.5)	0.69
Resection		32 (54%)	9 (39%)	23 (64%)	0.063
Endoscopic		34 (58%)	17 (74%)	17 (47%)	0.043
Stent		8 (16%)	6 (27%)	2 (7.4%)	0.12
Rings (n°)		2.2 (1.0)	1.7 (1.0)	2.6 (0.8)	0.001
1° Recurrence		12 (24%)	8 (35%)	4 (15%)	0.18
2° Recurrence		4 (8.3%)	2 (9.1%)	2 (7.7%)	>0.99

**Table 4 jpm-13-00729-t004:** Complications divided by group and classified according to te Clavien–Dindo classification of surgical complications.

Group	Patient	Type of Complication	Grade of Complication	Treatment
COVID-19	1	Bilateral vocal folds paralysis	III	Surgical tracheostomy
	2	Respiratory failure due to pneumonia	III	OTI
	3	c. difficile infection	II	Antibiotics
	4	Respiratory failure followed by kidney failure	III	OTI + Dialysis
	5	VAP	III	OTI + antibiotics
	6	Granuloma relapse	III	Substitution of T-tube
Non-COVID	1	Anastomotic dehiscence	III	Definitive tracheostomy
	2	Anastomosis not possible	III	Definitive tracheostomy
	3	Severe subglottic stenosis	III	T-tube placement

## Data Availability

Data can be asked to corresponding author.

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
