# Peer review of "Analysis of Risk Factors for Tracheal Stenosis Managed during COVID-19 Pandemic: A Retrospective, Case-Control Study from Two European Referral Centre"

_jpm, 2023, doi:10.3390/jpm13050729_

Round 1

Reviewer 1 Report

Congratulations to you on your novel study about the effects of COVID on tracheal stenosis. The manuscript was well written with acceptable language although there are some occasional mistakes in sentence structure and typos.

However, the main defect in this article is the selection of the control population (non-COVID patients). Since everyone knows prolonged intubation causes tracheal stenosis. In this article the length of intubation in COVID cases was two times as that of non-COVID. So you should choose non-COVID patients with similar length of intubation as control in order to reflect the effect of COVID.

The following are some other minor defects which should be revised for further submission.

(1) The last sentence in 'Introduction',Line 80-83, was wrongly written. Please rephrase.

(2) In Demographic data, last sentence 'his duration'---> 'its duration'

(3) thorough ---> through

(4) surgical treatment in Fig. 1 are shown---> in Fig. 1 shows

(5) no-COVID---> non-COVID.

(6) In the second ''one" patient

(7) In line 304, this measures---> what measures ?

(8) In line 347, whit---> with ?  have not knowledge---> have no knowledge

(9) In the tables, there are no units for the variables shown, days, years, etc.

Author Response

Comments and Suggestions for Authors

Congratulations to you on your novel study about the effects of COVID on tracheal stenosis. The manuscript was well written with acceptable language although there are some occasional mistakes in sentence structure and typos.

However, the main defect in this article is the selection of the control population (non-COVID patients). Since everyone knows prolonged intubation causes tracheal stenosis. In this article the length of intubation in COVID cases was two times as that of non-COVID. So you should choose non-COVID patients with similar length of intubation as control in order to reflect the effect of COVID.

Thank you for your interesting and relevant suggestion. Effectively the length of intubation between non-COVID and Covid patients were different but the aim of our study was not exclusively identified risks factors which specifically reflect the effect of covid in determining tracheal stenosis. We believe that this aim is not actionable with the limits of a retrospective study as ours. With this retrospective study we have only investigated and reported as the surgical activity of two referral centre in tracheal surgery has been modified during COVID and post-covid pandemic and how the complexity of covid tracheal stenosis is major if comparing radiological data of the two populations. Furthermore, we believe that your suggestions is very important and we have added a sentence in discussion which expresses your perplexity. However, we want anticipate you that we are now leading a prospective study involving all covid patients hospitalised in ICU of 7 centres in Lombardy that have been intubated more than 7 days to identify the percentage of patients that have developed tracheal stenosis and to study the risk factors that have occurred in determining it.  

 The following are some other minor defects which should be revised for further submission.

 (1) The last sentence in 'Introduction', Line 80-83, was wrongly written. Please rephrase.

Thank you very much for this suggestion. The sentence has been correct.

 (2) In Demographic data, last sentence 'his duration'---> 'its duration'

Thank you for this correction that we made on text.

(3) thorough ---> through

Thank you for this correction that we made on text.

 (4) surgical treatment in Fig. 1 are shown---> in Fig. 1 shows

Thank you for this correction that we made on text.

 (5) no-COVID---> non-COVID.

Thank you, all words “no-COVID” have been replaced with “non-COVID”

 (6) In the second ''one" patient

The mistakes have been corrected

 (7) In line 304, this measures---> what measures ?

The measure has been explained

(8) In line 347, whit---> with ?  have not knowledge---> have no knowledge

Thank you, the mistakes have been corrected.

 (9) In the tables, there are no units for the variables shown, days, years, etc.

Thank you for your suggestion. The units fir the variables have been added to table 1.

Reviewer 2 Report

Dear Editor and Authors,

Thank you for the opportunity to review this manuscript.

The manuscript describes the clinical, demographic and radiological characteristics, as well as  treatment options in both COVID and non-COVID patients with subglottic/tracheal stenosis (post orotracheal intubation or idiopathic), addressed to two specialized airway disease centers.

A strong point of this study is the relatively large number of enrolled patients (considering the low frequency of this complication) and the fact that the study was conducted in two specialized in airway disease centers, which offered the possibility for obtaining important information and well organized study protocol due to the hospital`s rigorous routine.

However, there are some  concerns that the Authors should adress in order to improve the manuscript:

Abstract

Authors must be consistent in using abbreviations from the beginning until the end of the article (e.g. only SARS-CoV-2 instead of SAR-CoV-2 and Sars-CoV-2).

You should avoid expressions like "COVID-related tracheal stenosis", used throughout the manuscript, but rather use "tracheal stenosis in COVID patients", because in this descriptive study, a causal relationship between SARS-CoV-2 infection and the occurrence of tracheal stenosis cannot be demonstrated.

Introduction

Row 82 is repeated in row 83.

Material and methods

Regarding demographic data, it is specified that data related to clinical features (which are not analysed/not found in the manuscript) and comorbidities (which are written very briefly - ex cadiological) were collected. The inclusion of clinical data and their analysis would give more value to the study and perhaps following the statistical correlations, risk factors for tracheal stenosis can be better outlined (as the Authors wrote in the discussions section).

Also, the number of patients who were intubated (both COVID and non-COVID) should be included, not just the length of intubation, considering that prolonged orotracheal intubation is a risk factor for tracheal stenosis.

The statistical analysis is very briefly described.

Results

The tables must be improved: the unit of measure is not written for the numerical variables; in Table 1, for tracheotomy, the order is shifted; also the abbreviation "No" has different meanings (No COVID and No=number?) - it must be clarified in the table`s legend ; in Table 2, the measurement units in the text do not correspond to those in the table; Table 4 shows a classification (II-III) of complications that must be detailed in the text or table legend to make it clearer for readers.

If there is data on how soon after intubation the tracheotomy was performed, it would be valuable to include it in Table 1.

Conclusion

It would be interesting to know how many tracheal stenosis were treated in these two referral centers during the pre-pandemic period (the same number of months).

Author Response

Reviewer 2

Dear Editor and Authors,

Thank you for the opportunity to review this manuscript.

The manuscript describes the clinical, demographic and radiological characteristics, as well as  treatment options in both COVID and non-COVID patients with subglottic/tracheal stenosis (post orotracheal intubation or idiopathic), addressed to two specialized airway disease centers.

A strong point of this study is the relatively large number of enrolled patients (considering the low frequency of this complication) and the fact that the study was conducted in two specialized in airway disease centers, which offered the possibility for obtaining important information and well-organized study protocol due to the hospital`s rigorous routine.

However, there are some concerns that the Authors should address in order to improve the manuscript:

Abstract

Authors must be consistent in using abbreviations from the beginning until the end of the article (e.g. only SARS-CoV-2 instead of SAR-CoV-2 and Sars-CoV-2).

Thank you very much, we have finally corrected the manuscript by using exclusively SARS-CoV-2.

You should avoid expressions like "COVID-related tracheal stenosis", used throughout the manuscript, but rather use "tracheal stenosis in COVID patients", because in this descriptive study, a causal relationship between SARS-CoV-2 infection and the occurrence of tracheal stenosis cannot be demonstrated.

Thank you very much, we have finally corrected the manuscript by replacing the expressions like “COVID-related stenosis” with the expressions “tracheal stenosis in COVID patients".

Introduction

Row 82 is repeated in row 83.

Thank you, we have reformulated the sentence.

Material and methods

Regarding demographic data, it is specified that data related to clinical features (which are not analysed/not found in the manuscript) and comorbidities (which are written very briefly - ex cadiological) were collected. The inclusion of clinical data and their analysis would give more value to the study and perhaps following the statistical correlations, risk factors for tracheal stenosis can be better outlined (as the Authors wrote in the discussions section).

Thank you very much for your suggestions. Data related to clinical features and comorbidity are effectively briefly reported even if they were analyzed. Unfortunately, the only clinical data having statistical significancy was the BMI. We have now added a sentences specifying that other clinical data were not statistically significative. 

Also, the number of patients who were intubated (both COVID and non-COVID) should be included, not just the length of intubation, considering that prolonged orotracheal intubation is a risk factor for tracheal stenosis.

Thank you very much for your suggestions. The fact that all included patients were previously intubated was wrongly omitted. We have now added a sentence specifying that all patients included in this study were intubated.

The statistical analysis is very briefly described.

Thank you very much for this comment.

We described in details the statistical analysis performed, including the specific statistical tests used for group comparison.

Results

The tables must be improved: the unit of measure is not written for the numerical variables; in Table 1, for tracheotomy, the order is shifted; also the abbreviation "No" has different meanings (No COVID and No=number?) - it must be clarified in the table`s legend ; in Table 2, the measurement units in the text do not correspond to those in the table; Table 4 shows a classification (II-III) of complications that must be detailed in the text or table legend to make it clearer for readers.

Thank you very much for your suggestions.

The unit of measure for the numerical variable are now inserted for both table 1 and 2.

The abbreviation “no” has been replaced with “not-performed” to avoid misunderstanding.

The classification of complications reported in table 4 has been added to the legend to make clearer the table understanding for readers.

If there is data on how soon after intubation the tracheotomy was performed, it would be valuable to include it in Table 1.

Unfortunately, we do not have these data

Conclusion

It would be interesting to know how many tracheal stenosis were treated in these two referral centers during the pre-pandemic period (the same number of months).

In our experience (Humanitas, Milan, Italy) we have experienced a doubling volume of patients affected of tracheal stenosis after COVID pandemic compared with pre-pandemic period. It was the same feedback from Avicenne, Paris.

We want sincerely thank you for your revisions suggestion that allow us to improve the quality of our manuscript.

Reviewer 3 Report

Congratulations on this interesting study and results.

My questions are as follows. 

1. What was the pathology of the resected rings? Did the inflammation reached the cartilage or it was in the mucosa only? 

2.  On the basis of your results, how many repeated endoscopic dilatation can be useful and successful before the decision of the resection? With other words, how after how many or how frequent attempts should the patients send for surgery? 

Author Response

Congratulations on this interesting study and results.

My questions are as follows.

  1. What was the pathology of the resected rings? Did the inflammation reached the cartilage or it was in the mucosa only?

Thank you for your questions.

We have identified two types of problems.

In patient that were previously tracheostomized the major problem was the breaking of tracheal ring

In patients which have developed tracheal stenosis after COVID infection that were not tracheostomized the major problems was the inflammatory status of the section of resected trachea. In this subgroup of patients, the inflammation was not uniform. In some patients it was exclusively mucosal, in others it was transmural. We are now performing a study to understand if the presence of transmural inflammation of the section of resected trachea can represent a risk factor to develop tracheal stenosis recurrence after surgery.

  1. On the basis of your results, how many repeated endoscopic dilatation can be useful and successful before the decision of the resection? With other words, how after how many or how frequent attempts should the patients send for surgery?

Thank you for this interesting question. In our opinion endoscopic dilatation can represent a useful tool to delayed surgery in symptomatic tracheal stenosis to avoid surgery during inflammation status. To delay surgery after 3-6 months of covid infection should be, in our opinion, the best management to take. We have added this suggestion in discussion section (line 364).

Round 2

Reviewer 1 Report

All comments answered  and errors corrected